# OpenReview forum: "Training-Free Safe Text Embedding Guidance for Text-to-Image Diffusion Models"
_NeurIPS.cc/2025/Conference — NeurIPS 2025 poster_

### Official Review · Reviewer_5G8U · 2025-06-30

**Clarity:** 3
**Significance:** 3
**Originality:** 3
**Rating:** 4
**Confidence:** 3

**Summary:**

This paper introduces Safe Text Embedding guidance (STG) as a safeguard mechanism against image outputs that involve undesired or inappropriate content, that is subjective to the user. The proposed method extends Safe Data Guidance (SDG), that targets the diffusion model weights in its optimization objective by only targeting the text embedding that is going to be fed to the model. Given a guidance function $g$ that assesses the image outputs $x_0$ based the desired image behavior, STG applies the optimization objective that SDG defines as updates to the text embedding representation that guides the generation. Addressing the fact that the updates to the diffusion model can result in disturbed model behavior, STG only targets the embeddings and claim to preserve the generation properties. Experiments on concepts such as nudity, violence demonstrate the effectiveness of the method across competing approaches.

**Questions:**

- Does the method generalize to generative models that uses multiple text encoders? This is an important concern as it determines the types of generative models that the proposed approach can be applied to.
- Does the proposed method be applied to tasks such as bias mitigation? In addition to unpleasant concepts like violent content and nudity, biased representations such as gender imbalances across occupations is also a problem that safety methods address (like UCE). It is an important clarification that if STG can be extended to such tasks.

**Ethical Concerns:**

["NO or VERY MINOR ethics concerns only"]

**Final Justification:**

The questions that I had towards the computational efficiency, generalizability to newer models (compared to Stable Diffusion), application to bias mitigation have been addressed by the supplementary experiments provided as a part of the rebuttal. Furthermore, I  adjusted my score as leaning towards positive.

**Limitations:**

- The paper does not have an explicit discussion on the limitations. However, since the work explicitly addresses the embedding space rather than the overall model, a limitations section on the concepts that can be mitigated and which cannot be mitigated would benefit the paper, to determine its applicability. The overall societal impact of the paper is positive, where discussions on the broader impact is provided in the supplementary material.

**Paper Formatting Concerns:**

The paper is in suitable format.

**Quality:**

3

**Strengths And Weaknesses:**

**Strengths**
- The paper proposes an extension of SDG that promises improved preservation properties of the properties of the model.
- The proposed method demonstrates its effectiveness in preserving model behavior while preventing the generation of the undesired concepts (e.g. nudity)
- The paper is well experimented, where the motivation is clear as well as the quantitative improvements.
- Technical details are sufficiently explained, where the claims are coupled with proofs.

**Weaknesses**
- The GPU consumption of the method is significant which limits the applicability of the method to larger models.
- The method addresses the source of undesired concepts to be determined in the embedding space. This narrows the applicability of the method to tasks such as bias removal, that is also related with the cross-attention and self-attention weights.
-  The proposed method serves as an incremental update on top of SDG. While it is significant that the model properties are preserved better, as only the embedding space is targeted, the technical novelty of the work is questionable and can be considered as an incremental update on top of the previous methods.
- The method requires updates on the text embeddings fed into the generative model. Does the method generalize to approaches that uses more than one text embeddings for generation, as such approaches are more frequently used (FLUX, SDXL, SD3)?

---

> ### Author Rebuttal · Authors · 2025-07-31
>
> We appreciate the reviewers’ constructive and encouraging comments. We have carefully considered all suggestions and provide our responses below.
> ***
> > ### **W1: [GPU consumption]**
>
> We appreciate the reviewer’s observation regarding GPU consumption. As discussed in Appendix C.3 and D, the added inference cost of STG mainly stems from gradient computations required for updating the text embedding. To mitigate this overhead, we explore time- and memory-efficient techniques. Specifically, we apply half-precision (FP16) inference during sampling, which significantly reduces both runtime and GPU memory usage, while preserving the safety performance of our method, as shown in the below table. We extended our runtime and memory usage analysis (originally presented in Table 7 of the appendix) by including results with half-precision (FP16) inference. Although STG requires gradient computation during update steps, we observe that FP16 inference maintains nearly the same safety performance. This suggests that existing time- and memory-efficient techniques can be effectively leveraged to address the computational cost of STG in practice.
>
> **[Table D1: Sampling time and memory usage with FP16]**
>
> |Method|FP16|Time (s/batch)|GPU memory (GB)|DSR (↑)|PP (↑)|
> |----------------------------|----|----------------|-----------------|------|------|
> |Base|X|15.8|8.23|0.08|-|
> |SLD|X|22.7|13.5|0.76|0.65|
> |SAFREE|X|23.2|13.6|0.36|0.73|
> |STG (ours)||||||
> |$\rho=2.0$, 15% update|X|23.9|45.4|0.79|0.90|
> |$\rho=2.0$, 15% update|O|14.0|22.8|0.79|0.89|
> |$\rho=2.0$, 40% update|X|30.4|45.4|0.88|0.84|
> |$\rho=2.0$, 40% update|O|20.7|22.8|0.92|0.84|
> |$\rho=0.5$, 80% update|X|50.8|45.4|0.92|0.84|
> |$\rho=0.5$, 80% update|O|35.0|22.8|0.91|0.84|
>
> ***
> > ### **W2 & Q2: [Application of bias mitigation]**
>
> Thank you for the suggestion. STG can be extended to bias mitigation tasks if a suitable safety function can be defined to measure and discourage biased behavior. To explore this, we conduct a preliminary experiment targeting gender imbalance across occupations. Specifically, we use the prompt format ‘a photo of {occupation}’ and analyze the gender distribution of generated images from the Stable Diffusion v1.4 backbone. We observe that prompts such as ‘nurse’ resulted in 100\% female images, while ‘farmer’ yielded 98% male images, indicating strong gender bias in the base model.
>
> To mitigate this, we define a safety function as the negative squared difference between CLIP scores for ‘a photo of male {occupation}’ and ‘a photo of female {occupation}’. This formulation encourages the generated image to avoid being overly aligned with either gender-specific prompt, thereby guiding it toward a more neutral representation. As shown in the below table, the degree of bias mitigation can be controlled by adjusting the update scale hyperparameter $\rho$. The reported ratio corresponds to the proportion of male-presenting images, and values closer to 0.5 indicate that the generated images are achieving a more balanced gender distribution, demonstrating effective mitigation of occupational gender bias.
>
> **[Table D2: Gender distribution in generated images for occupation prompts]**
>
> |Occupation|Method|$\rho$|# Man|# Woman|Ratio|
> |---|---|---|---|---|---|
> |Nurse|Base||0|250|0|
> ||STG|0.5|43|207|0.172|
> |||1.0|58|192|0.232|
> |||1.5|117|133|0.468|
> ||
> |Farmer|Base||246|4|0.984|
> ||STG|0.5|171|79|0.684|
> |||1.0|167|83|0.668|
> |||1.5|152|98|0.608|
>
> That being said, we note that STG operates on a per-instance basis and does not explicitly enforce balance over the distribution of generated samples. As such, it may produce images that appear more neutral, but this does not necessarily indicate fair distribution-level behavior. We view this as a potential direction for future work, particularly in integrating group-level fairness constraints into our framework.
>
> ***
> > ### **W3: [Incremental update on SDG]**
>
> While the proposed method may appear to be an incremental modification of SDG, shifting the guidance target from the image latent space to the text embedding space; our work provides a thorough theoretical and empirical justification for this change. Specifically, Section 4.3 with Theorem 1 analytically shows that updating the text embedding implicitly influences the latent space while preserving the underlying model likelihood. Section 4.4 further supports this claim with toy experiments, demonstrating that STG more faithfully maintains the original generative distribution compared to SDG, especially when the safety function is imperfect. This distinction is particularly important for tasks such as style removal or copyright mitigation, where it is critical to preserve the semantic content or visual fidelity of the original image while modifying only specific attributes.
>
> In addition, although SDG itself is derived from the previous Universal Guidance framework, our work is the first to adapt and implement both SDG and STG for the safety domain, along with extensive evaluations across safety tasks. We believe this comprehensive adaptation and evaluation further contributes to the field.
>
> ***
>
> > ### **W4 & Q1: [Multiple text embeddings]**
>
> Our method is compatible with recent generative models that utilize multiple text embeddings, such as FLUX, SDXL, and SD3. In these models, STG can be applied by updating all text embeddings in the same way. We conduct additional experiments using more recent diffusion models, including FLUX, SDXL, and SD3. For each model, we follow the default sampling and configuration settings provided in the diffusers library. In the case of FLUX, we use the guidance-distilled version.
>
> The below table reports the results on the Ring-A-Bell (violence) benchmark for safe generation metrics, and COCO dataset for general generation quality. Note that for this table, the COCO evaluation is conducted using 1,000 images, which accounts for the scale difference in FID values.
>
> **[Table D3: Results on recent diffusion models: FLUX, SDXL, and SD3]**
>
> |Models|Methods|Ring-A-Bell DSR (↑)|Ring-A-Bell PP (↑)|COCO FID-1k (↓)|COCO CLIP (↑)|
> |-----|-------------------------|---------------------|--------------------|--------------|---------------|
> |FLUX|Base|0.1120|-|56.58|0.3267|
> ||STG (ours, $\tau$=0.20)|0.2800|0.8527|56.59|0.3267|
> ||STG (ours, $\tau$=0.18)|0.5280|0.7031|56.52|0.3260|
> ||STG (ours, $\tau$=0.16)|0.6960|0.5997|57.77|0.3200|
> ||
> |SDXL|Base|0.0360|-|48.97|0.3380|
> ||STG (ours, $\tau$=0.20)|0.2520|0.8788|49.24|0.3378|
> ||STG (ours, $\tau$=0.18)|0.5040|0.7991|49.11|0.3366|
> ||STG (ours, $\tau$=0.16)|0.7720|0.7191|49.44|0.3302|
> ||
> |SD3|Base|0.1160|-|53.70|0.3344|
> ||STG (ours, $\tau$=0.20)|0.4200|0.7562|53.89|0.3342|
> ||STG (ours, $\tau$=0.18)|0.5400|0.6652|53.65|0.3333|
> ||STG (ours, $\tau$=0.16)|0.6760|0.5678|54.91|0.3293|
>
> As shown in the table, the Ring-A-Bell prompts still induce harmful outputs even in the recent models, as indicated by the low DSR values for the base models. Applying STG improves safety (higher DSR), while maintaining similar image quality as measured by FID and CLIP score. These results demonstrate that STG generalizes well across diverse backbones, pipelines, and fast generation methods. Additionally, the trade-off between image quality and safety can be adjusted via the update threshold hyperparameter $\tau$, providing flexibility depending on application requirements.

---

> ### Author Response · Authors · 2025-08-06
>
> Dear Reviewer 5G8U,
>
> We sincerely appreciate your valuable feedback and constructive comments. We have tried to address your comments in our rebuttal, and we would be grateful to know whether our responses have resolved your concerns. If there are any remaining questions or points to discuss, we would be happy to discuss them further. Thank you again for your time and thoughtful review.
>
> Best regards,
> Authors

---

> ### Comment · Reviewer_5G8U · 2025-08-06
> **Thanks for the Rebuttal**
>
> Thanks to the authors for the rebuttal. My primary concern in my review was the generalizability of the method to newer models, computational efficiency and the application of the method towards tasks such as bias mitigation overall. My questions have been addressed in the author rebuttal. Given that the other reviewers are also leaning towards the acceptance of the paper, and the supplementary experiments presented in the rebuttal, I will be adjusting my score towards positive. The authors are strongly encouraged to include these supplementary examples in the camera-ready version. Also I believe qualitative samples over models such as FLUX, SD3 and SDXL would also benefit the paper in a positive way.

---

> > ### Author Response · Authors · 2025-08-06
> >
> > We appreciate the constructive feedback. We're pleased that the rebuttal helped clarify the concerns. As the reviewer suggested, we will include the supplementary examples in the camera-ready version. Thank you again for the time and effort dedicated to the review.

---

### Official Review · Reviewer_ypdn · 2025-06-30

**Clarity:** 3
**Significance:** 3
**Originality:** 3
**Rating:** 4
**Confidence:** 3

**Summary:**

The paper proposes a training-free method for text-to-image diffusion models and introduces safe text embedding guidance (STG) to dynamically adjust the text embedding during sampling to suppress unsafe content. At each diffusion step, STG computes the safety score on the expected denoised image and uses the first-order Taylor approximation to show the equivalent to injecting a “safe guidance” term into the score function. Extensive experiments demonstrate that STG consistently outperforms both training-based and training-free baselines.

**Questions:**

1. What about the safety performance on more categories?
2. What about the generalization on other models?

**Ethical Concerns:**

["NO or VERY MINOR ethics concerns only"]

**Final Justification:**

My recommended score is 4, Borderline accept

**Limitations:**

See the above weakness.

**Paper Formatting Concerns:**

No issues

**Quality:**

3

**Strengths And Weaknesses:**

Strengths:
1. Zero-training operates entirely at inference time, avoiding costly retraining or curated datasets.
2. Theoretical derivation via Tweedie’s formula and explicit decomposition into original score + safe guidance provides clear interpretability.

Weakness:
1. The method effectiveness depends on the accuracy and bias of the external classifier.
2. The gradient computations and safety evaluations at each sampling step reduces the efficiency.
3. What about the generalization on other models?

---

> ### Author Rebuttal · Authors · 2025-07-31
>
> We thank the reviewer for the helpful feedback. Our responses to the raised concerns are summarized below.
> ***
>
> > ### **W1: [Reliance of the external classifier]**
>
> We agree with the reviewer’s concerns, and this limitation is explicitly acknowledged in Appendix D of the paper. However, as discussed in Sections 4.3 and 4.4, our theoretical analysis and the toy experiments demonstrate that STG could be effective when the safety function is not a perfect estimator of the true safety probability. Unlike data-space guidance methods, STG jointly considers the model likelihood and safety objective, which helps mitigate the risk of overfitting to potentially biased or imperfect proxy metrics.
>
> Recent advances in pre-trained vision-language models, such as CLIP, provide a promising alternative. These models enable flexible zero-shot classification and can be leveraged to construct effective proxy safety functions. In our experiments, the violence and artist-style removal tasks utilized CLIP, even though CLIP was not specifically designed for those safety objectives.
>
> Together, our theoretical and empirical findings suggest that STG could be robust to the safety functions and can effectively improve safety while preserving the model’s original generative capabilities. This supports the practical use of proxy classifiers, such as CLIP-based scores, within our framework.
>
>
> ***
>
> > ### **W2: [Reduced efficiency]**
>
> We appreciate the reviewer’s observation regarding the efficiency. As discussed in Appendix C.3 and D, the added inference cost of STG mainly stems from gradient computations required for updating the text embedding. To mitigate this overhead, we explore time- and memory-efficient techniques. Specifically, we apply half-precision (FP16) inference during sampling, which significantly reduces both runtime and GPU memory usage, while preserving the safety performance of our method, as shown in the below table. We extended our runtime and memory usage analysis (originally presented in Table 7 of the appendix) by including results with half-precision (FP16) inference. Although STG requires gradient computation during update steps, we observe that FP16 inference maintains nearly the same safety performance. This suggests that existing time- and memory-efficient techniques can be effectively leveraged to address the computational cost of STG in practice.
>
> **[Table C1: Sampling time and memory usage with FP16]**
>
> |Method|FP16|Time (s/batch)|GPU memory (GB)|DSR (↑)|PP (↑)|
> |----------------------------|----|----------------|-----------------|------|------|
> |Base|X|15.8|8.23|0.08|-|
> |SLD|X|22.7|13.5|0.76|0.65|
> |SAFREE|X|23.2|13.6|0.36|0.73|
> |STG (ours)||||||
> |$\rho=2.0$, 15% update|X|23.9|45.4|0.79|0.90|
> |$\rho=2.0$, 15% update|O|14.0|22.8|0.79|0.89|
> |$\rho=2.0$, 40% update|X|30.4|45.4|0.88|0.84|
> |$\rho=2.0$, 40% update|O|20.7|22.8|0.92|0.84|
> |$\rho=0.5$, 80% update|X|50.8|45.4|0.92|0.84|
> |$\rho=0.5$, 80% update|O|35.0|22.8|0.91|0.84|
>
> Additionally, we would like to highlight an observation from Table 4 in the main text. By adjusting the update step ratio, under comparable sampling time, STG attains a similar defense success rate to SLD while achieving substantially higher safety performance than SAFREE. Also, it does so with better prior preservation than both methods. This demonstrates that STG offers flexible trade-offs between sampling speed, safety, and fidelity, depending on the desired deployment scenario.
>
> ***
>
> > ### **W3 & Q2: [Generalization]**
>
> To evaluate the generalization ability of STG, we conducted experiments using the PixArt-alpha model, a recent diffusion model with a Transformer-based backbone, in combination with the DPM-Solver sampler. These results are reported in Table 6 and discussed in Appendix C.2.
>
> Additionally, to further address concerns about scalability, we extend our experiments to include more recent diffusion models such as FLUX, SDXL, and SD3. For each model, we follow the default sampling and configuration settings provided in the diffusers library. In the case of FLUX, we use the guidance-distilled version.
>
> The below table reports the results on the Ring-A-Bell (violence) benchmark for safe generation metrics, and COCO dataset for general generation quality. Note that for this table, the COCO evaluation is conducted using 1,000 images, which accounts for the scale difference in FID values.
>
> **[Table C2: Results on recent diffusion models: FLUX, SDXL, and SD3]**
>
> |Models|Methods|Ring-A-Bell DSR (↑)|Ring-A-Bell PP (↑)|COCO FID-1k (↓)|COCO CLIP (↑)|
> |-----|-------------------------|---------------------|--------------------|--------------|---------------|
> |FLUX|Base|0.1120|-|56.58|0.3267|
> ||STG (ours, $\tau$=0.20)|0.2800|0.8527|56.59|0.3267|
> ||STG (ours, $\tau$=0.18)|0.5280|0.7031|56.52|0.3260|
> ||STG (ours, $\tau$=0.16)|0.6960|0.5997|57.77|0.3200|
> |SDXL|Base|0.0360|-|48.97|0.3380|
> ||STG (ours, $\tau$=0.20)|0.2520|0.8788|49.24|0.3378|
> ||STG (ours, $\tau$=0.18)|0.5040|0.7991|49.11|0.3366|
> ||STG (ours, $\tau$=0.16)|0.7720|0.7191|49.44|0.3302|
> |SD3|Base|0.1160|-|53.70|0.3344|
> ||STG (ours, $\tau$=0.20)|0.4200|0.7562|53.89|0.3342|
> ||STG (ours, $\tau$=0.18)|0.5400|0.6652|53.65|0.3333|
> ||STG (ours, $\tau$=0.16)|0.6760|0.5678|54.91|0.3293|
>
>
> As shown in the table, the Ring-A-Bell prompts still induce harmful outputs even in the recent models, as indicated by the low DSR values for the base models. Applying STG improves safety (higher DSR), while maintaining similar image quality as measured by FID and CLIP score. These results demonstrate that STG generalizes well across diverse backbones, pipelines, and fast generation methods. Additionally, the trade-off between image quality and safety can be adjusted via the update threshold hyperparameter $\tau$, providing flexibility depending on application requirements.
>
> We also investigate sensitivity of STG to different samplers by conducting experiments on the Stable Diffusion v1.4 backbone using the DDPM sampler, keeping all settings identical to those used with the DDIM sampler (as in Figure 3a for the Ring-A-Bell nudity task). Due to the rebuttal policy, we are only able to provide tabular results at this stage. For methods originally visualized as curves, we report three representative points based on DSR (the lowest, median, the highest values). We will include the full curves in the revised version of the paper.
>
> **[Table C3: Results with DDPM samplers]**
>
> |                | Methods         | DSR (↑) | PP (↑) |
> |----------------|-----------------|---------|--------|
> |                | Base            | 0.5260  | -      |
> | Training-based | ESD             | 0.4421  | 0.6431 |
> |                | SPM             | 0.6632  | 0.5290 |
> |                | DUO (min)       | 0.5895  | 0.7817 |
> |                | DUO (medium)    | 0.9474  | 0.6844 |
> |                | DUO (max)       | 0.9789  | 0.4701 |
> | Training-free  | Negative Prompt | 0.2842  | 0.6808 |
> |                | UCE             | 0.6105  | 0.6101 |
> |                | RECE            | 0.8421  | 0.5209 |
> |                | SLD-Medium      | 0.2211  | 0.7773 |
> |                | SLD-Strong      | 0.7684  | 0.6639 |
> |                | SLD-Maximum     | 0.9053  | 0.4980 |
> |                | SAFREE          | 0.3053  | 0.6774 |
> |                | SDG (min)       | 0.1158  | 0.7979 |
> |                | SDG (medium)    | 0.4526  | 0.7611 |
> |                | SDG (max)       | 0.9368  | 0.6143 |
> |                | STG (min)       | 0.8632  | 0.7508 |
> |                | STG (medium)    | 0.9789  | 0.7210 |
> |                | STG (max)       | 1.0000  | 0.5022 |
>
> The results show that STG consistently delivers better performance across different samplers, indicating that STG is robust to the sampler.
>
> Across these diverse backbones and samplers, STG consistently demonstrates strong performance, highlighting its generality and robustness across a wide range of model architectures and sampling strategies.
>
> ***
>
> > ### **Q1: [More categories]**
>
> We understand the reviewer’s question as referring to safety performance across a broader range of inappropriate categories (e.g., hate, harassment, etc.). To investigate this, we conduct experiments on the I2P benchmark, which includes prompts from seven categories used in the SLD paper: hate, harassment, violence, self-harm, sexual, shocking, and illegal activity.
>
> Due to the large number of prompts in I2P benchmark, we adopt a fast diffusion model, specifically the Latent Consistency Model (LCM) (Luo et al., 2023) with 4 sampling steps to enable efficient sampling. For the safety function $g$, we use the CLIP score between each image and the fixed general inappropriate text prompt, as defined in the SLD paper. This unified safety function allows STG to be applied consistently across all seven categories. As commonly done in the I2P dataset, we compute the DSR using the combined Q16/NudeNet classifier and report the experimental results.
>
> **[Table C4: Safety performance (DSR) across diverse categories on the I2P benchmark using LCM + STG]**
> |         | hate   | harassment | violence | self-harm | sexual | shocking | illegal activity | * | COCO FID | COCO CLIP |
> |-------------------|--------|------------|----------|-----------|--------|----------|------------------|---|----------|-----------|
> | Fixed             | 0.5628 | 0.6663     | 0.6151   | 0.6017    | 0.7368 | 0.5210   | 0.6191           | * | 36.20    | 0.3019    |
> | STG ($\rho$=0.16) | 0.6494 | 0.7318     | 0.7037   | 0.6742    | 0.7744 | 0.6192   | 0.6862           | * | 37.66    | 0.2923    |
> | STG ($\rho$=0.15) | 0.7403 | 0.8058     | 0.7553   | 0.7141    | 0.8174 | 0.6951   | 0.7261           | * | 41.21    | 0.2779    |
>
> As shown in the table, STG demonstrates strong safety performance even when applied to modern diffusion models and across diverse safety categories.
>
> (Luo et al., 2023) Latent Consistency Models: Synthesizing High-Resolution Images with Few-Step Inference

---

> ### Comment · Reviewer_ypdn · 2025-08-07
> **Official Comment**
>
> Thanks for your detailed rebuttal. My concerns are resolved and I keep my score.

---

> > ### Author Response · Authors · 2025-08-07
> >
> > Thank you for the response. We're glad to hear that your concerns have been resolved. We sincerely appreciate your time and effort throughout the review process.

---

### Official Review · Reviewer_QgQT · 2025-07-02

**Clarity:** 2
**Significance:** 3
**Originality:** 3
**Rating:** 4
**Confidence:** 4

**Summary:**

To address the problem of unsafe and undesired generations of T2I models, the paper proposes a training-free guidance approach that steers a pre-trained model using a signal from an external safety classifier. Specifically, the method builds on the Universal Guidance framework and defines a time-dependent safety function using classifier feedback over intermediate denoised images. The core idea is to adjust the text embedding at each diffusion step to nudge the generation toward safer outputs. The authors evaluate their approach on three tasks, namely nudity, violence, and artist-style removal on Stable Diffusion v1.4, and compare against both training-free and training-based safety methods.

**Questions:**

- In Figure 3, several methods (e.g., SLD, SAFREE, STG) are represented by multiple data points. Could you clarify what these points correspond to? Do they reflect different hyperparameter settings (τ, ρ), in case of STG, or variants in case of SLD (e.g., SLD-M, SLD-S, SLD-Max)?
- Can you explain the gap in reported values for SAFREE between your results and the original paper (e.g., ASR of 0.114 vs. DSR of 0.36)?
- For the violence detection task, could you elaborate on why GPT-4o was chosen and whether its performance was validated in this specific context?

**Ethical Concerns:**

["NO or VERY MINOR ethics concerns only"]

**Final Justification:**

Overall, the rebuttal’s content and the promised revisions strengthen the paper; therefore, I lean towards acceptance. The clarifications on hyperparameter settings, baseline configurations, and dataset differences address my main concerns about the experimental setup. The added evaluations with alternative metrics are particularly helpful and reveal the classifier dependency as an important limitation (e.g., NudeNet/Falconsai). However, this limitation, alongside the computational overhead, restricts the applicability of the approach in broader or real-time settings. Nonetheless, the insights and framework proposed here could serve as a valuable foundation for future work on improving training-free safety guidance methods.

**Limitations:**

Limited discussion. Next to the computational overhead, compared to other approaches, the introduced methods rely on "safety functions" which may reduce flexibility.

**Quality:**

2

**Strengths And Weaknesses:**

**Strengths**

- The proposed method is well-motivated and clearly introduced, offering a lightweight solution that does not require model fine-tuning.
- Comprehensive comparisons to current state-of-the-art training-free (e.g., SLD, SAFREE) and training-based (e.g., ESD, DUO) methods are provided.
- Ablations analyze the impact of key hyperparameters such as the step ratio (ρ) and threshold (τ).
- Further, the authors present that their approach can be combined with training-based methods, resulting in further enhanced safety.

**Weaknesses**

- Unlike SLD or SAFREE, which make use of textual representations of undesired content to guide the safe generation, the proposed method requires an external classifier, reducing its generality and flexibility. Unfortunately, this dependency is not discussed as a limitation.

- Overall, the limitations of the method, besides computational overhead, are not well addressed.

- The experimental setup lacks clarity. While the authors state that the strength of guidance is controlled via the hyperparameters ρ and τ, the specific values used in the experiments and how they were selected are not detailed. Similarly, hyperparameter settings for baseline methods are not well described. For instance, it is only mentioned in the appendix that the SLD variant SLD-M is used, which is the weakest configuration the original paper reports. Further reported DSR values for SAFREE do not match the original reported values. Specifically, the original SAFREE paper reports an ASR (Attack Success Rate) of 0.114 on the Ring-A-Bell benchmark, while the present paper reports a DSR (1-ASR) of 0.36.

- Unclear suitability of evaluation metrics: For nudity, the same classifier is used both to guide generation and to evaluate output safety, raising questions about metric bias. For violence, GPT-4o is used, but its reliability in detecting violent content is unclear and not further investigated.

---

> ### Author Rebuttal · Authors · 2025-07-31
>
> We greatly appreciate the reviewers' insightful comments and provide our responses below.
> ***
> > ### **W1 & W2: [Requirement of external classifier]**
>
> As the reviewer pointed out, Our method relies on an external classifier for the safety function. This is feasible for tasks with public classifiers (e.g., NudeNet for nudity) but limits generality where none exist. However, recent advances in pre-trained vision-language models, such as CLIP, provide a promising alternative. These models enable flexible zero-shot classification and can be leveraged to construct effective proxy safety functions. In our experiments, the violence and artist-style removal tasks utilized CLIP, even though CLIP was not specifically designed for those safety objectives.
>
> This explanation of the safety function is briefly mentioned in Section 4.1 (lines 129-131), but we agree that the reliance on external classifiers and implications for generality should be explicitly stated as a limitation. We will revise the manuscript accordingly to include this discussion.
>
> Additionally, as discussed in Section 4.3 and 4.4, our theoretical analysis and the toy experiments demonstrate that STG could be effective when the safety function is not a perfect estimator of the true safety probability. Unlike data-space guidance methods, STG jointly considers the model likelihood and safety objective. This allows STG to better preserve the original generative capabilities while improving safety, which supports the practical use of proxy classifiers (e.g., CLIP scores) within our framework.
>
> ***
> > ### **W3 & Q2: [Experimental setup]**
>
> **(Hyperparameter settings)**
>
> We provided the specific hyperparameter values of STG in Appendix B.1 (lines 717-735). As illustrated in our results (e.g., Figure 3 in the main text), there is a trade-off between the defense success rate and prior preservation. To reflect this, we report multiple variants across a range of hyperparameter values to illustrate this balance.
>
> We set $\gamma$ based on sampling time constraints, as it directly determines the number of update steps during sampling. We usually set to $\gamma=0.6$. The threshold $\tau$ defines which samples are considered unsafe (and thus require updates), and is inherently tied to the range and semantics of the safety function $g$. For example, in the artist-style removal task, where $g$ is defined as the CLIP score difference between ‘art’ and a target artist, the decision boundary is expected near zero, so $\tau$ is chosen close to zero accordingly. The update scale $\rho$ controls the trade-off between safety and prior preservation, and can be adjusted flexibly at inference time depending on the desired balance. Since our method is training-free and modular, $\rho$ can be easily adapted for different use cases or safety definitions without any retraining overhead.
>
> For baseline methods, their respective experimental configurations are provided in Appendix B.2 (lines 737-776), where we describe the setup for each method used in our comparisons.
>
> **(SLD variants)**
>
> We clarify that the main results (Figure 3 in the main text) include all three SLD variants: Max, Strong, Medium. These are represented by the green solid line, ordered from left to right as Max, Strong, Medium.
> While SLD-Medium yields the lowest safety performance among the three configurations, it achieves the best performance in terms of preserving the original generative capabilities, particularly in terms of prior preservation and general generation metrics such as FID and CLIP score on COCO. For this reason, we chose SLD-Medium as the representative configuration in Table 2, which focuses on general generation quality. We also report the results for SLD-Strong and SLD-Max in the below table:
>
> **[Table B1: SLD-Strong and SLD-Max results for Table 2 in the main paper (COCO evaluation)]**
> |Method|FID(↓)|CLIP(↑)|
> |---|---|---|
> |Base|23.22|31.96|
> |...|
> |SLD-Medium|24.32|31.29|
> |SLD-Strong|25.10|30.33|
> |SLD-Max|27.69|29.28|
> |...|
> |STG (ours)|22.00|31.14|
>
> **[Table B2: SLD-Strong and SLD-Max results for Table 3 in the main paper (artist-style removal)]**
> |Method|Remove|"Van| Gogh"||*|Remove|"Kelly| McKernan"||
> |---|---|---|---|---|---|---|---|---|---|
> ||LPIPS_e↑|LPIPS_u↓|ACC_e↓|ACC_u↑|*| LPIPS_e↑|LPIPS_u↓|ACC_e↓|ACC_u↑|
> |||
> |Base|||1.00|0.89|*|||0.90|0.71|
> |...|||||*|||||
> |SLD-Medium|0.28|0.12|0.60|0.81|*|0.22|0.18|0.50|0.74|
> |SLD-Strong|0.39|0.19|0.40|0.79|*|0.32|0.28|0.30|0.68|
> |SLD-Max|0.50|0.34|0.65|0.58|*|0.49|0.40|0.10|0.59|
> |...|||||*|||||
> |STG (ours)|0.46|0.08|0.30|0.85|*|0.58|0.10|0.10|0.65|
>
> As expected, FID and CLIP score on the COCO deteriorate for SLD-Strong and SLD-Max. For the artist-style removal task, SLD-Max improves removal on the erased concepts but unintentionally affects unerased concepts.
>
> **(SAFREE reported values)**
>
> The discrepancy in reported DSR for SAFREE originates from different Ring-A-Bell benchmark versions. As noted in the second issue thread on SAFREE’s official Github repository, SAFREE used a custom set from RECE authors, which we used the official adversarial prompts (later released by the Ring-A-Bell authors) following DUO, as mentioned in Appendix B.1 (lines 687-688).
> A comparison of the two datasets reveals significant differences. The version used in SAFREE consists of human-readable prompts, while the official version we used contains highly adversarial prompts that are much harder to interpret. This distinction explains the observed discrepancy in performance.
>
> We also evaluate STG on SAFREE’s version and reproduce their results; STG still shows superior performance. We will revise the manuscript to clearly specify datasets and configurations.
>
> **[Table B3: Results on Ring-A-Bell (nudity) used in SAFREE]**
> |Method|ASR (↓)|PP (↑)|
> |---|---|---|
> |SAFREE (reproduced)|0.1013|0.5367|
> |STG (ours)|**0.0886**|**0.5820**|
>
> ***
> > ### **W4 & Q3: [Evaluation metric]**
>
> We appreciate the reviewer’s concern regarding potential metric bias in our evaluation protocol.
>
> For the nudity task, we acknowledge that using the same classifier (NudeNet) for both generation guidance and evaluation may introduce bias. To address this, we additionally evaluate the generated images using an alternative metric: an open-source ViT-based NSFW image classifier provided by Falcons.ai. We present a comparison between the original DSR values obtained using the NudeNet detector and those computed using the Falconsai classifier for each point in Figure 3a. While there are some mismatches and variations between the two metrics, we observe that STG consistently achieved higher DSR at similar levels of Prior Preservation (PP). Furthermore, as shown in Table 2 of the main paper, STG also maintains general generation quality.
>
> **[Table B4: DSR results using Falconsai classifier]**
> ||Methods|PP (↑)|DSR-NudeNet (↑)|DSR-Falconsai (↑)|
> |---|---|---|---|---|
> ||Base|-|0.0842|0.3263|
> |Training-based|ESD|0.6509|0.6000|0.7684|
> ||SPM|0.5389|0.7053|0.8947|
> ||DUO (min)|0.8958|0.4842|0.3368|
> ||DUO (medium)|0.7210|0.9684|0.6526|
> ||DUO (max)|0.4747|0.9895|0.9579|
> |Training-free|Negative Prompt|0.7323|0.3579|0.6842|
> ||UCE|0.6418|0.6842|0.7895|
> ||RECE|0.5400|0.9263|0.9684|
> ||SLD-Medium|0.8708|0.2526|0.6632|
> ||SLD-Strong|0.6515|0.7579|0.9579|
> ||SLD-Maximum|0.4829|0.9474|1.000|
> ||SAFREE|0.7242|0.3579|0.3263|
> ||SDG (min)|0.9140|0.4737|0.3789|
> ||SDG (medium)|0.7574|0.7158|0.5789|
> ||SDG (max)|0.6055|1.0000|0.9684|
> ||STG (min)|0.9238|0.6632|0.6316|
> ||STG (medium)|0.8446|0.9158|0.7368|
> ||STG (max)|0.6280|0.9895|0.9368|
>
> Regarding violence, we agree with the reviewer’s concern about the reliability of GPT-4o as an evaluation tool. In our work, we followed the evaluation protocol introduced by DUO, which also used GPT-4o for safety assessment for violence.
>
> To further validate this metric, we additionally evaluate our results using the Q16 classifier, an open-source model commonly used for detecting harmful content. Specifically, we collect pairs of safety assessment scores from both GPT-4o and the Q16 classifier across various model and hyperparameter configurations. Then, we compute the Pearson correlation between these scores, which resulted in a value of 0.943. This strong linear correlation suggests that GPT-4o provides safety evaluations that are highly consistent with those of an established classifier like Q16.
>
> We will include the Q16-based evaluation curves and this discussion in the revised version.
>
> ***
> > ### **Q1: [Configurations in Figure 3]**
>
> As mentioned in our response to W3, the SLD results in Figure 3 are represented by the green solid line, with the three variants, SLD-Max, Strong, and Medium, ordered from left to right.
>
> For STG and SDG, the multiple points correspond to different hyperparameter settings, as detailed in Appendix B.1 (lines 726-732) for STG and Appendix B.2 (lines 758-762) for SDG.
> Specifically, for STG on the nudity task, the points represent different (ρ, τ) combinations: {(1.8, 0.01), (1.3, 0.01), (0.5, 0.01), (0.5, 0.03), (0.5, 0.2)}, plotted from left to right in Figure 3. For the violence task, τ is fixed at 0.05, and ρ is varied across {3, 2, 1, 0.5, 0.2, 0.1}, also in left-to-right order. For SDG, in the nudity task, τ is fixed at 0.01 and ρ is varied as {5, 1, 0.7, 0.5, 0.1}; for the violence task, τ is fixed at 0.1 and ρ is varied as {40, 15, 10, 5, 1}. Again, in both cases, the points correspond to the values listed from left to right.
>
> For DUO, the blue line in Figure 3 corresponds to different values of the hyperparameter β, varied as {100, 250, 500, 1000, 2000}, in left-to-right order. These settings are described in Appendix B.2 (lines 768-770).
>
> Note that SAFREE is shown as a single point in Figure 3. For this method, we adopted the default hyperparameter configuration provided in the official SAFREE code, as noted in Appendix B.2 (lines 742-744).
>
> We will explicitly include these details in the revised version of the paper to ensure clarity and avoid confusion for readers.

---

> ### Comment · Reviewer_QgQT · 2025-08-05
>
> Thank you for the detailed rebuttal and additional experiments. The clarifications on hyperparameter settings, baseline configurations, and dataset differences address my main concerns about the experimental setup. The added evaluations with alternative safety metric are particularly helpful and reveal the classifier dependency as an important limitation (nudenet/falcsonsai), which should be stated in the final version. Overall, the rebuttal strengthens the paper, and I encourage explicitly including the new evaluation results and the limitation discussion in the final version.

---

> > ### Author Response · Authors · 2025-08-05
> >
> > Thank you for the thoughtful feedback. We're glad the rebuttal addressed the concerns, and the comments were helpful in improving the paper. As the reviewer suggested, we will explicitly include the new evaluation results and the limitation discussion in the final version. We appreciate the time and effort spent on the review.

---

### Official Review · Reviewer_YpFz · 2025-07-03

**Clarity:** 4
**Significance:** 4
**Originality:** 3
**Rating:** 4
**Confidence:** 3

**Summary:**

This paper addresses the safety concerns of text-to-image (T2I) diffusion models, which are often trained on large-scale web-crawled datasets and thus susceptible to generating biased or harmful content, especially when prompted with malicious inputs. To mitigate this, the authors propose Safe Text Embedding Guidance (STG), a training-free method that adjusts text embeddings at inference time using a safety function. Theoretically, the method aligns the model’s output distribution with safety constraints while minimizing degradation in image quality. Empirical results demonstrate that STG outperforms both training-based and training-free baselines across various safety scenarios, showing strong performance in terms of safety, alignment, and generation quality.

**Questions:**

1) The proposed method is closely tied to the denoising process. How sensitive is STG to the choice of sampler? For example, would it behave differently under schedulers?
2) Is STG compatible with recent fast-generation methods such as Latent Consistency Models (LCMs)? If not, what are the main challenges in adapting it to such frameworks?
3) Most experiments are conducted on older models and samplers. Could the authors provide more evidence or discussion on how STG would scale to more recent architectures and pipelines?
4) Table 4 appears to contain a directional error in the PP column (arrow direction). Could the authors clarify or correct this?
5) While the paper acknowledges the increased sampling time, this remains a critical limitation. Are there any strategies to mitigate this cost except for update frequency?
If the authors can provide convincing responses to the concerns above, especially regarding scalability to modern architectures and compatibility with fast generation methods, I would be open to increasing my score.

**Ethical Concerns:**

["NO or VERY MINOR ethics concerns only"]

**Final Justification:**

Most of my concerns were addressed in the rebuttal. My initial assessment was borderline accept, and while it remains borderline, I am now leaning more toward acceptance.

**Limitations:**

Yes

**Paper Formatting Concerns:**

No formatting concerns.

**Quality:**

3

**Strengths And Weaknesses:**

Strengths:

1) The paper is well-structured and clearly written, making it accessible even to readers unfamiliar with safety in generative models.
2) The motivation is well-grounded, and the proposed method is logically and theoretically supported.
3) Despite being training-free, STG achieves competitive or superior performance compared to more complex training-based methods, which is a notable contribution to the community.
4) The experimental section is comprehensive, including ablations and hyperparameter sensitivity analyses that strengthen the empirical claims.

Weaknesses:

1) The scalability of the method is somewhat unclear. Most experiments are conducted on relatively outdated settings (e.g., Stable Diffusion 1.4 with DDIM). Although the appendix includes results on PixArt-α and DPM-Solver, these are still not state-of-the-art.
2) While the paper acknowledges its limitations, the increased sampling time introduced by STG is a significant drawback.

---

> ### Author Rebuttal · Authors · 2025-07-31
>
> We sincerely appreciate the reviewers’ careful assessment and constructive feedback. Below, we provide detailed responses to each comment.
> ***
>
> > ### **W1 & Q3: [Scalability]**
>
> Thank you for the insightful comment. We acknowledge that using Stable Diffusion v1.4 with DDIM sampler may be considered relatively outdated. We selected this setting primarily to ensure fair comparison with prior safety-focused works, many of which report results using this same configuration.
>
> To address the concern regarding scalability, we conduct additional experiments using more recent diffusion models, including FLUX, SDXL, and SD3. For each model, we follow the default sampling and configuration settings provided in the diffusers library. In the case of FLUX, we use the guidance-distilled version.
>
> The below table reports the results on the Ring-A-Bell (violence) benchmark for safe generation metrics, and COCO dataset for general generation quality. Note that for this table, the COCO evaluation is conducted using 1,000 images, which accounts for the scale difference in FID values.
>
> **[Table A1: Results on recent diffusion models: FLUX, SDXL, and SD3]**
>
> | Models | Methods                 | Ring-A-Bell DSR (↑) | Ring-A-Bell PP (↑) | COCO FID-1k (↓) | COCO CLIP (↑) |
> |--------|-------------------------|---------------------|--------------------|--------------|---------------|
> | FLUX   | Base                    | 0.1120              | -                  | 56.58             |    0.3267          |
> |        | STG (ours, $\tau$=0.20) | 0.2800              | 0.8527             |  56.59        |    0.3267           |
> |        | STG (ours, $\tau$=0.18) | 0.5280              | 0.7031             |     56.52    | 0.3260              |
> |        | STG (ours, $\tau$=0.16) | 0.6960              | 0.5997             |       57.77       |   0.3200            |
> ||
> | SDXL   | Base                    | 0.0360               | -                  | 48.97             |      0.3380         |
> |        | STG (ours, $\tau$=0.20) |      0.2520    |   0.8788         |   49.24        |   0.3378            |
> |        | STG (ours, $\tau$=0.18) |      0.5040       |      0.7991    |   49.11      | 0.3366              |
> |        | STG (ours, $\tau$=0.16) |      0.7720         |    0.7191         |  49.44     | 0.3302              |
> ||
> | SD3    | Base                    | 0.1160              | -                  | 53.70        | 0.3344       |
> |        | STG (ours, $\tau$=0.20) | 0.4200              | 0.7562             |    53.89   |0.3342               |
> |        | STG (ours, $\tau$=0.18) | 0.5400              | 0.6652             |      53.65        |       0.3333        |
> |        | STG (ours, $\tau$=0.16) | 0.6760              | 0.5678             |   54.91           |      0.3293         |
>
> As shown in the table, the Ring-A-Bell prompts still induce harmful outputs even in the recent models, as indicated by the low DSR values for the base models. Applying STG improves safety (higher DSR), while maintaining similar image quality as measured by FID and CLIP score. These results demonstrate that STG generalizes well across diverse backbones, pipelines, and fast generation methods. Additionally, the trade-off between image quality and safety can be adjusted via the update threshold hyperparameter $\tau$, providing flexibility depending on application requirements.
>
> ***
>
> > ### **W2 & Q5: [Increased sampling time]**
>
> We appreciate the reviewer’s observation regarding the increased sampling time. As discussed in Appendix C.3 and D, the added inference cost of STG mainly stems from gradient computations required for updating the text embeddings. To mitigate this overhead, we explore time- and memory-efficient techniques. Specifically, we apply half-precision (FP16) inference during sampling, which significantly reduces both runtime and GPU memory usage, while preserving the safety performance of our method, as shown in the below table. We extended our runtime and memory usage analysis (originally presented in Table 7 of the appendix) by including results with half-precision (FP16) inference. Although STG requires gradient computation during update steps, we observe that FP16 inference maintains nearly the same safety performance. This suggests that existing time- and memory-efficient techniques can be effectively leveraged to address the computational cost of STG in practice.
>
> **[Table A2: Sampling time and memory usage with FP16]**
>
> | Method                 |  FP16   | Time (s/batch) | GPU memory (GB) | DSR (↑)  | PP (↑)   |
> |----------------------------|----|----------------|-----------------|------|------|
> | Base                        |  X  | 15.8           | 8.23            | 0.08 | -    |
> | SLD                         |  X  | 22.7           | 13.5            | 0.76 | 0.65 |
> | SAFREE                      | X  | 23.2           | 13.6            | 0.36 | 0.73 |
> | STG (ours)                   |   |                |                 |      |      |
> |   $\rho=2.0$, 15% update      | X  | 23.9           | 45.4            | 0.79 | 0.90 |
> |   $\rho=2.0$, 15% update  | O | 14.0           | 22.8            | 0.79 | 0.89 |
> |   $\rho=2.0$, 40% update      | X | 30.4           | 45.4            | 0.88 | 0.84 |
> |   $\rho=2.0$, 40% update  | O | 20.7           | 22.8            | 0.92 | 0.84 |
> |   $\rho=0.5$, 80% update      |  X | 50.8           | 45.4            | 0.92 | 0.84 |
> |   $\rho=0.5$, 80% update  | O | 35.0           | 22.8            | 0.91 | 0.84 |
>
> Additionally, we would like to highlight an observation from Table 4 in the main text. By adjusting the update step ratio, under comparable sampling time, STG attains a similar defense success rate to SLD while achieving substantially higher safety performance than SAFREE. Also, it does so with better prior preservation than both methods. This demonstrates that STG offers flexible trade-offs between sampling speed, safety, and fidelity, depending on the desired deployment scenario.
>
> ***
> > ### **Q1: [Different sampler]**
>
> Thank you for the thoughtful question. To evaluate how sensitive STG is to the choice of sampler, we conduct additional experiments using the Stable Diffusion v1.4 backbone with the DDPM sampler, keeping all settings identical to those used with the DDIM sampler (as in Figure 3a for the Ring-A-Bell nudity task). Due to the rebuttal policy, we are only able to provide tabular results at this stage. For methods originally visualized as curves, we report three representative points based on DSR (the lowest, median, the highest values). We will include the full curves in the revised version of the paper.
>
> **[Table A3: Results with DDPM samplers]**
>
> |                | Methods         | DSR (↑) | PP (↑) |
> |----------------|-----------------|---------|--------|
> |                | Base            | 0.5260  | -      |
> | Training-based | ESD             | 0.4421  | 0.6431 |
> |                | SPM             | 0.6632  | 0.5290 |
> |                | DUO (min)       | 0.5895  | 0.7817 |
> |                | DUO (medium)    | 0.9474  | 0.6844 |
> |                | DUO (max)       | 0.9789  | 0.4701 |
> | Training-free  | Negative Prompt | 0.2842  | 0.6808 |
> |                | UCE             | 0.6105  | 0.6101 |
> |                | RECE            | 0.8421  | 0.5209 |
> |                | SLD-Medium      | 0.2211  | 0.7773 |
> |                | SLD-Strong      | 0.7684  | 0.6639 |
> |                | SLD-Maximum     | 0.9053  | 0.4980 |
> |                | SAFREE          | 0.3053  | 0.6774 |
> |                | SDG (min)       | 0.1158  | 0.7979 |
> |                | SDG (medium)    | 0.4526  | 0.7611 |
> |                | SDG (max)       | 0.9368  | 0.6143 |
> |                | STG (min)       | 0.8632  | 0.7508 |
> |                | STG (medium)    | 0.9789  | 0.7210 |
> |                | STG (max)       | 1.0000  | 0.5022 |
>
> The results show that STG consistently delivers better performance across different samplers, indicating that STG is robust to the sampler.
>
> ***
> > ### **Q2: [Compatible with fast generation methods]**
>
> STG can be integrated with various fast generation methods. For example, in Table 6 of Appendix C.2, we demonstrate successful application of STG with the DPM-Solver sampler, which is a representative fast generation method of the sampler side, on the PixArt-alpha backbone.
>
> Our method only requires access to the mean predicted images $\bar{x}_0$ at intermediate diffusion timesteps during sampling. Since most diffusion models and samplers, including LCMs, provide this, STG can generally be applied without architectural modifications.
>
> We also conduct experiments with Latent Consistency Models on Ring-A-Bell (violence). We find that STG is compatible with this framework as well.
>
> **[Table A4: Results on the fast generation method: LCMs]**
>
> | Methods                 | Ring-A-Bell DSR (↑) | Ring-A-Bell PP (↑) | COCO FID-3K (↓) | COCO CLIP (↑) |
> |-------------------------|---------------------|--------------------|--------------|---------------|
> | Base                    | 0.0200              | -                  | 36.20        | 0.3019        |
> | STG (ours, $\tau$=0.20) | 0.2280              | 0.6576             | 36.22        | 0.3013        |
> | STG (ours, $\tau$=0.18) | 0.5160              | 0.5867             | 36.28        | 0.3006        |
> | STG (ours, $\tau$=0.16) | 0.8040              | 0.4806             | 37.66        | 0.2923        |
>
> ***
>
> > ### **Q4: [Correction of directional error in table]**
>
> Thank you for pointing that out. The arrow direction in the PP column should be an upward arrow ($\uparrow$). PP (Prior Preservation) is a metric where higher values are better. We will make this correction in the revision.

---

> > ### Comment · Reviewer_YpFz · 2025-08-05
> >
> > Thank you for the detailed rebuttal. I have carefully reviewed your response, and I appreciate the clarification regarding the scale. My concern has been addressed, so I will maintain my positive score.

---

> > > ### Author Response · Authors · 2025-08-05
> > >
> > > Thank you for taking the time to review our rebuttal. We’re glad to hear that your concern has been addressed, and we appreciate the continued positive evaluation.

---

### Note · Authors · 2025-08-12

We sincerely appreciate all reviewers for their constructive and thoughtful feedback throughout the review process.

We are encouraged by the recognition that our proposed method is well-motivated (*YpFz, QgQT*), logically and theoretically supported (*YpFz, ypdn, 5G8U*), effective despite being training-free (*YpFz, ypdn, 5G8U*), that the experimental section is comprehensive (*YpFz, QgQT, 5G8U*), and that combining STG with training-based methods leads to further enhancement (*QgQT*).

We are also pleased that the concerns raised were addressed in our rebuttal. In particular, we provided new experiments and analyses that:
* **Generalizability** (*YpFz, ypdn, 5G8U*)
  * We demonstrated that STG scales well to recent diffusion models with multiple text embeddings (FLUX, SDXL, SD3) **[Table A1 in YpFz W1 & Q3]**, fast-generation methods (Latent Consistency Models) **[Table A4 in YpFz Q2]**, and the different samplers (DDPM) **[Table A3 in YpFz Q1]**.
  * Consistent with the main text, which demonstrated STG’s generalizability via artist-style removal (Table 3), we extended the evaluation to broader scenarios in the rebuttal, including the I2P benchmark across multiple inappropriate categories **[Table C4 in ypdn Q1]** and bias mitigation for occupational gender imbalance **[Table D2 in 5G8U W2 & Q2]**.
* **Efficiency** (*YpFz, ypdn, 5G8U*)
  * We reduced sampling time and memory usage via time- and memory-efficient techniques such as half-precision inference, which preserved safety performance while reducing computational cost **[Table A2 in YpFz W2 & Q5]**.
* **External classifier reliance** (*QgQT, ypdn*)
  * We showed both theoretically (Section 4.3) and empirically (using CLIP scores not tailored to specific safety) that STG remains effective with proxy safety functions. In addition, as illustrated in **[Table B4 in QgQT W4 & Q3]**, alternative classifiers produce consistent results. We note that image-based safety detection methods continue to evolve and improve, and our framework is inherently adaptable to integrate such advances.
* **Experimental setup clarity** (*QgQT*)
  * We clarified hyperparameter settings, baseline configurations, and dataset differences, as discussed in **[QgQT W3 & Q2, Q1]**.

We appreciate the reviewers’ acknowledgement that these additions strengthened the paper, and we will incorporate the new results, clarifications, and discussions into the revised version.

---

### Decision · Program_Chairs · 2025-09-17

**Decision:**

Accept (poster)

**Comment:**

**Summary:** This paper presents Safe Text embedding Guidance (STG), a training-free framework for improving safety in text-to-image diffusion by directly modifying text embedding guidance. STG employs a safety function to detect unsafe semantic directions and steer embeddings away from them during generation. Theoretical analysis shows that aligning perturbed data with both the model distribution and safety constraints within classifier-free guidance reduces unsafe outputs while preserving semantic alignment. Experiments on unsafe prompts demonstrate substantial safety improvements with minimal impact on image quality.

**Strengths:** Key strengths noted were: (1) the simplicity and practicality of STG; (2) the novelty of training-free embedding-level steering with a safety function; (3) the theoretical insight on how classifier-free guidance aligns embeddings with safety constraints; (4) comprehensive experiments with ablations, and (5) the strong balance between safety and fidelity.

**Weaknesses:** Initial concerns were: (1) unclear scalability and limited evaluation on newer diffusion models (e.g., SDXL, SD3, FLUX); (2) efficiency issues from added gradient computations, slower sampling, and high GPU use; (3) dependence on an external classifier, introducing potential bias and reduced generality; (4) unclear experimental setup with under-specified hyperparameters and questionable baseline choices; and (5) limited novelty, as STG may be seen as an incremental extension of SDG focused on embeddings.

During the rebuttal, the authors clarified design choices, addressed scalability issues by providing experiments with recent diffusion frameworks, provided justifications on increased sampling time and improved baseline comparisons. They emphasized STG’s practicality as a lightweight, training-free complement to more heavyweight alignment methods. While concerns about dependency on specific classifier remain, reviewers agreed that the contribution is novel, timely, and impactful.